# Accelerometric Assessment of Postural Balance in Children: A Systematic Review

**DOI:** 10.3390/diagnostics11010008

**Published:** 2020-12-22

**Authors:** Jose L. García-Soidán, Raquel Leirós-Rodríguez, Vicente Romo-Pérez, Jesús García-Liñeira

**Affiliations:** 1Special Didactics Department, Faculty of Education and Sport Sciences, Universidade de Vigo, Campus a Xunqueira, s/n. 36005 Pontevedra, Spain; jlsoidan@uvigo.es (J.L.G.-S.); jesgarcia@alumnos.uvigo.es (J.G.-L.); 2Nursing and Physical Therapy Department, Faculty of Health Sciences, Universidad de León, Ave. Astorga, 15, 24401 Ponferrada, Spain; 3Didactics and School Organization and Research Methods Department, Faculty of Education and Sport Sciences, Universidade de Vigo, Campus a Xunqueira, s/n. 36005 Pontevedra, Spain; vicente@uvigo.es

**Keywords:** pediatrics, motor control, balance, posture, evaluation, child development

## Abstract

The correct development of postural control in children is fundamental to ensure that they fully reach their psychomotor capacities. However, this capacity is one of the least studied in the clinical and academic scope regarding children. The objective of this study was to analyze the degree of implementation of accelerometry as an evaluation technique for postural control in children and how it is being used. Methods: A systematic search was conducted in PubMed, SpringerLink, SportsDiscus, Medline, Scopus, and Web of Science with the following terms: balance, postural control, children, kids, accelerometry, and accelerometer. Results: The search generated a total of 18 articles. Two groups of studies were differentiated: those which exclusively included healthy individuals (*n* = 5) and those which included children with pathologies (*n* = 13). Accelerometry is being used in children mainly to assess the gait and static balance, as well as to identify the differences between healthy children and children with developmental disorders. Conclusions: Accelerometry has a discrete degree of implementation as an evaluation tool to assess postural control. It is necessary to define a systematic method for the evaluation of postural control in pediatrics, in order to delve into the development of this capacity and its alterations in different neurodevelopmental disorders.

## 1. Introduction

The correct development of postural control (PC) in children is fundamental and necessary to ensure that they fully reach their psychomotor capacities [1]. This includes the development of dynamic balance and static postural control, which allows a child to interact with the environment in an independent manner and with the lowest possible risk of injury or falling [2]. The development of PC, although it is present from the first moments of a person’s life, occurs especially at the age of 6–10 years [3]. This period of maturation is due to the fact that adult-like PC strategies appear around the age of 7–8 years [4]. These PC strategies are characterized by the optimization of the coordination of movements between the head and the trunk [3,5,6], and by a change in the way in which the brain controls and manages visual, somatosensory, and vestibular inputs, based on feedback [7,8]. All this coincides, in the same period, with the development of other important maturation phenomena in the central nervous system and the acquisition of other complex motor abilities [9].

However, this physical capacity is one of the least studied in the clinical and academic scope regarding children [10]. This is partly due to the fact that the evaluation of PC includes the assessment of different components and aptitudes, such as postural stability, coordination, muscular strength, center of mass control, anticipatory and reactive neuromuscular reactions, motor control, the correct reception of proprioceptive, visual, and vestibular stimuli, and, finally, the correct processing and management of all these signals in the central nervous system for the development of efficient motor responses [6,11,12,13]. The reliable and valid evaluation of all the processes and subsystems that participate in PC is very complex and hinders the development of evaluation and diagnostic tests [14]. In Pediatric clinical practice, balance assessments are usually based on qualitative methods, which are inefficient and have low reliability and sensitivity [15]. Reliable tests have been developed, but they require the use of expensive force or pressure platforms, magnetic tracking, infrared emitter, electronic pressure-sensitive walkway, or surface electromyographic recordings for the determination of the individual’s center of pressure, an indicator of great clinical validity, reliability, and sensitivity to identify relatively premature sensory–motor deficits [16,17].

In the last decade, the analysis of movement through wearable sensors, such as accelerometers (ACCs), became popular [18]. Among the wide range of accelerometers, triaxial ACCs are very light portable devices (a few grams in weight) that record the accelerations of the body (or the body segment to which they are attached) in the three axes of space [14]. That is, they record the quality of such movement as a function of the changes in its velocity (acceleration), although they do not quantify them (degrees or range of movement performed). These properties have made ACCs the research instruments of choice for the development of reliable, sensitive, and economical evaluation tools that can be used in clinical and healthcare environments in specialties such as neurology and geriatrics [14,18,19].

Previous studies have evaluated balance with ACCs (especially with the aim of analyzing the center of body mass) [16], but the majority of published studies have focused on adult or elderly populations and have placed great emphasis on the risk of falls [20,21,22,23,24]. Accelerometric balance assessment has been repeatedly compared with other quantitative and qualitative balance assessment methods with positive results in different populations (elderly with a history of falls or post-stroke, patients with Huntington’s or Parkinson’s disease, and those with Friedreich’s ataxia with vestibular disorder) [14]. In reference to the child population, several authors have reported difficulty in the use of ACCs for collecting data for short periods of activity (indispensable for the assessment of individuals at a young age), because the equilibrium reactions of children are characterized as vigorous and producing “bursts” [25,26].

Therefore, the main objective of this literature review is to show the current degree of implementation of accelerometry as an evaluation technique for PC in children and how it is being used (functional tests performed, parameters recorded, and data analyses conducted), based on the hypothesis that accelerometric evaluation allows the state of PC to be assessed in a quantitative manner and with greater sensitivity compared with other evaluation techniques.

## 2. Materials and Methods

### 2.1. Search Strategy

This study was registered on PROSPERO and followed the Preferred Reporting Items for Systematic Reviews and Meta-analyses (PRISMA) reporting guidelines ant the recommendations from the Cochrane Collaboration [27,28]. The PICO question was then chosen as follows: P—population: children; I—intervention: postural control assessment with accelerometers; C—control: traditional balance assessment tests in the clinical setting; O—outcome: accelerometric variables; S—study designs: experimental and descriptive studies. A systematic search of publications was conducted throughout the month of May 2020 in the following databases: PubMed, SpringerLink, SportsDiscus, Medline, Scopus, and Web of Science. The search strategy included different combinations with the following terms: balance, postural control, children, kids, accelerometry, and accelerometer.

From the studies found in the literature search, this review included those that were (a) written in English and (b) published in the last five years (2016 to present), (c) which used ACCs to evaluate postural control (both static and dynamic), and (d) whose sample included children aged 6–12 years among their participants. On the other hand, this investigation excluded (a) studies with neither experimental nor observational methodology (systematic reviews, editorials, etc.) and (b) studies that used ACCs to exclusively quantify the levels of physical activity. The search and selection process is shown in Figure 1. The full search strategy is available from the authors on request.

### 2.2. Data Extraction and Analysis

The following data were extracted from the selected articles: aim of the study, characteristics of the sample (age, inclusion and exclusion criteria, and number of participants), postural control evaluation tests used, characteristics of the ACCs employed (model, previous settings, and localization of the device for the recording of the measurements), post-processing of the accelerometric data (data handling performed and variables used for the analysis), and results obtained.

The Oxford 2011 Levels of Evidence, the Grading of Recommendations, Assessment, Development and Evaluation (GRADE) system and Evidence Alert Traffic Light System are not applicable in this case because what is evaluated is an assessment method. Instead of an analysis of the obtained results, the methodology of the selected studies was analyzed, following the method previously applied by Papi et al. [29]. These authors quantitatively evaluated investigations that applied wearables for the analysis of the movements of the human body based on the description of the objectives, the description of the design and methodology of the study, the choice of population and description of the sample, the description and preparation of the measurement instruments, the placing and fastening of the sensor, the sampling signals and frequency used, the data processing, the statistical methods employed, the comparison with gold standards, the main findings described, and the limitations. Each item was valued with 0 points (not described), 1 point (limited description), or 2 points (adequately described).

## 3. Results

The search and selection process generated a total of 18 articles that used ACC for the evaluation of PC in children. The methodological characteristics of each of the analyzed studies are shown in Table 1, Table 2, Table 3, Table 4, Table 5 and Table 6. Furthermore, the quality of the selected articles was established under the following scale: low (<33.3%), medium (33.4–66.7%), and high (>66.8%) [29,30]. Of the 18 articles analyzed, 7 showed medium quality [31,32,33,34,35,36,37] and 10 showed high quality [38,39,40,41,42,43,44,45,46,47] (Appendix A).

### 3.1. Experimental Methodology and Application Objectives

The objectives of the analyzed studies were varied. Those which employed a cross-sectional observational methodology were focused on estimating the normative assessment data through the Sway Balance System (SBS) method [31], evaluating the magnitude of anticipatory postural adjustments [42] and quantifying accelerations during gait (during normal gait [33,37], with trajectory and velocity changes [39], and while performing a cognitive task [35]). One study combined the analysis of static balance and balance during gait in order to determine the reliability and validity of accelerometry to evaluate balance in children [46]. Other cross-sectional studies used ACCs for the analysis of static balance with the aim of estimating the normative assessment data through the Balance Error Scoring System (BESS) method [41], evaluating PC of children in a seated position [44], determining the relationship between PC during gait and PC in a seated position [36], and assessing the effect of regular physical activity practice on the development of PC [47]. Finally, the rest of the cross-sectional studies determined the role of the limbs in the PC of the trunk [43], compared static balance between children with typical development (TD) and children with development coordination disorder (DCD) [45], and evaluated the applicability of the Huple device as an instrument for the rehabilitation of children with different pathologies [48]. A blinded, observational, case-control study was focused on evaluating static and dynamic balance in children with bilateral cochleo-vestibular loss with cochlear implants [35].

Regarding the investigations that included an intervention, two of them used ACCs to quantify accelerations during gait in the different spatial axes, with the aim of evaluating the efficacy of patient-tailored rehabilitation programs (through an experimental, controlled, randomized, single-blinded study) [40] and assessing hypotherapy (through a case series study) [32]. Lastly, an experimental, controlled, randomized study evaluated the improvement of postural control after an intervention of neuromuscular training [38].

### 3.2. Characteristics of the Samples

Two groups of studies were differentiated based on the following aspect: those which exclusively used heathy individuals in their samples [31,39,41,46,47] and those which included children with pathologies. Among the latter, the state of postural control in children with cerebral palsy (CP) was the study object in eight studies: exclusively in four of them [32,36,43,44], using data of children with TD as the control group in three [37,40,42], and, lastly, analyzing children with TD, healthy young and older adults, and older people with subacute stroke in one study [34]. Balance was also studied in children with DCD [33,38,45], bilateral cochleo-vestibular loss [35], and different neurodevelopmental disorders [48].

The samples used were small, with less than 70 participants in most of the cases [32,33,35,36,37,39,40,42,43,44,46,48]. Only six studies analyzed larger samples [31,34,38,41,45,47], with sample sizes ranging from 85 participants (of whom only 25 were children) [34] to 6762 participants (of whom only 606 were children) [41].

With respect to the sex distribution of the samples used, most of the studies employed samples mainly composed of males [31,32,33,35,36,37,38,41,43,44,45] (especially the case of Cheng et al. [38], who only included 16% of girls in their sample). In contrast, the study of Shiratori et al. [42] used a female proportion of 70%. Finally, seven studies used relatively balanced samples in terms of sex distribution [34,37,39,40,46,47,48], with female proportions between 50% [46,47] and 45% [37].

### 3.3. ACCs Used and Their Application

Different measurement wearables with ACCs incorporated were used. Triaxial ACCs were the most frequent option [31,32,33,34,35,36,38,39,41,43,44,45,46,47,48], with a predominant tendency to use accelerometric devices included in iOS mobile devices (iPhone^®^, iPod Touch^®^ o iPad^®^, Apple Inc., Cupertino, California, USA) [31,35,41]. On the other hand, three studies employed inertial units (MIMUs, Opal, APDM Inc., Portland, Oregon, USA [37,40] and x-IMU, x-io Technologies Limited, Bristol, UK [35]) and uniaxial ACCs (PCB Piezoelectronics, Depew, NY, USA) [42].

The settings used for the realization of the accelerometric recordings varied between 50 Hz [46,47] and 1000 Hz [42], although the most employed frequency was 100 Hz [32,34,36,39,41]. Moreover, it is important to point out that this measurement parameter was not described in five of the analyzed studies [31,35,38,40,45].

Regarding the localization of the sensors, these were placed mostly in the lumbo-pelvic region: in the iliac crest [39,41], at the level of the L2–L3 [32,34,36,43,44], L4 [46,47] and L5–S1 vertebrae [37,40], or simply indicating that it was placed on the mid-lumbar region [39]. Three studies performed the accelerometric measurement from the head (from its vertex [35,37] or from the occipital bone [40]), and five studies conducted it from the center of the chest or sternum [31,33,37,40,45]. Lastly, the hands of the participants were also employed as recording points [38,42,43] using the Huple device [48].

After the recordings and prior to the statistical analysis, several studies processed the accelerometric data through the application of low-pass Butterworth filters (between 3.5 Hz [41] and 20 Hz [34,37,40,42]), threshold filters to eliminate the noise of the signal [46], and attenuation coefficients [40]. None of the rest of the articles described the treatment of the data prior to the statistical analysis [31,32,33,35,36,38,39,43,44,45,48].

Finally, the variables used for the statistical analysis were, most frequently, the identification of the average accelerometric values and their root mean square (RMS), both in static balance [36,43,44,45,46,47] and during gait [33,34,37,39,40,46,47]. In particular, Brett et al. [31] quantified the magnitude of the postural changes using their own algorithm, with a value range of 0 to 100. Linder et al. [41] analyzed the angular and linear accelerations, applying the normalized path length, with this value representing the sum of the differences in the accelerations of the center of mass in all directions and converting these to Z-scores (a positive Z value indicates that balance is worse than the established normative mean). Shiratori et al. [42] analyzed the anticipatory postural adjustments by evaluating the acceleration of the dominant hand at the time it begins to hold a load, with the first detected acceleration being related to the anticipatory muscular activation (complementarily measured with electromiography). Separately, the estimation of the time of muscular latency, measured with electromiography as well, was also used as a study variable by Fong et al. [45].

Three studies calculated spatio-temporal variables related to the gait (such as the frequency and length of the steps and speed) [32,36,40], and one study estimated the local dynamic stability, applying the Lyapunov exponent [33].

### 3.4. Clinical Tests Conducted and Comparison with Gold Standard Tests

For the realization of the accelerometric recordings, some studies performed sets of functional tests that had been previously validated: the Pediatric Balance Scale (PBS) [34], the Gross Motor Function Measure [34], the BESS [41], the Sway Sport Balance System [31] (which were employed as gold standards to evaluate accelerometry), the Modified Clinical Test Sensory Interaction in Balance [35], the Movement Assessment Battery for Children [33,45], the Bruininks–Oseretsky Test of Motor Proficiency-2 [35], the Trunk Impairment Scale (TIS) [36,44] and the Trunk Control Measurement Scale (TCMS) [36,43,44]. Lastly, the Quality of Upper Extremity Skills Test, the Jebsen Taylor Hand Function Test, the Abilhand-KIDS, the Sitting Assessment Test for Children with Neuromotor Dysfunction, and the Box and Blocks Test were only used by Kim et al. [43,44].

Moreover, there were studies that based their evaluation on functional tests such as walking [32,33,39,40,46,47] and maintaining static balance (bipodal or monopodal) [42,46,47]. In other studies, accelerometry was complementarily used with other tests, such as computerized dynamic posturography [38,45] and electromyography [42,45].

To evaluate and contrast the accelerometric evaluation, the video recording of the tests [35,36,37,40], photoelectric cells for the analysis of the gait [36], and strength platforms [42] were used as gold standards.

### 3.5. Results Obtained

Two studies compared their results using the BESS, a functional balance test that had already been validated and referenced. This test produces errors during the recordings. However, accelerometry prevents the evaluator from making measuring errors, which has previously been questioned. The use of the SBS [31] and the Cleveland Clinic Postural Stability Index [41] has been reliably correlated with the results obtained through accelerometric devices (r = 0.8; *p* < 0.001).

Another functional test with which accelerometry has been compared is the Berg Balance Scale and its version for children, i.e., the PBS [34]. The latter has 14 items related to daily balance activities specifically designed for children. Tramontano et al. [34] used the Timed Up and Go and the Tinetti Balance Scale, two functional balance tests that have been widely validated for the measurement of balance in older people or individuals with pathologies. After the analysis of the results, the authors did not detect significant differences in the comparison of the accelerometric results of the PBS with those of the Berg Balance Scale between participants with CP and adult individuals after stroke (R^2^ = 0.056, *p* = 0.3). However, they did not compare this pediatric scale between healthy children and children with CP.

Saether et al. [36] compared the existing correlation between the results of the accelerometric analysis and those of the TIS and TCMS tests. The TIS was developed to evaluate trunk control in adults after stroke, by assessing the static and dynamic sitting balance and trunk coordination in sitting position [49]. The total score consisted of the sum of the three subscale scores, ranging from 0 (lowest performance) to 23 (best performance). The TIS has been tested for reliability and validity in children and adolescents with CP in the age group of 5 to 18 years [50,51]. The TCMS was developed from the TIS and expanded to include assessments of selective trunk movements and dynamic reaching. The total score ranged from 0 (lowest performance) to 58 (best performance), where the total score was the sum of three subscale scores: static sitting balance, dynamic sitting balance-selective movement control, and dynamic sitting balance-reaching [52]. The TCMS has been tested for reliability and validity in children with CP in the age group of 8 to 15 years [53]. Both Saether et al. [36] and Kim et al. [43,44] found significant correlations between the TCMS and TIS tests and the accelerometric measurements (0.6 < r > 0.8; *p* < 0.006; for the three).

The use of strength platforms and functional tests was applied by Shiratori et al. [42], who combined the use of the OR-6 strength platform (AMTI, Watertown, NY, USA) with accelerometry and the electromyography (EMG) test. Furthermore, all this was combined with the digitalized signals of the three systems using the personalized LabView software (USA Instruments, Aurora, OH, USA), which stores all the data for their subsequent processing. In this case, the ACC only analyzed the movement of the dominant hand, without finding significant results (F = 0.45, *p* = 0.6). That is, it was not used as an indirect measurement of the postural control of the whole body.

Computerized dynamic posturography (Smart Equitest; NeuroCom International Inc, Clackamas, OR, USA) was another method used in one study [38]. This system generates the SES (sway energy score), which is a nondimensional score in the acceleration velocity of the center of mass. In this case, the authors also used the ACC to quantify only the movement of the hands (which held the device) without finding significant results (*p* = 0.64). Thus, they did not measure the postural control of the whole body.

The video analysis of the gait was also used by Iosa et al. [40] and Summa et al. [37], who conducted the video recording of the tests using commercial video cameras (JVC GC-PX10, HD Memory) to calculate spatio-temporal parameters of the gait, such as the number, frequency, and length of the steps. In both cases, they obtained as result that the accelerations are significantly different between children with TD and CP both in the sternum, as well as in the head and pelvis (*p* < 0.008, in the three cases) [37], and between the acceleration attenuation coefficient from pelvis to head (*p* = 0.048) and from pelvis to sternum (*p* = 0.021) [40].

## 4. Discussion

The aim of this review was to show the current degree of implementation of accelerometry as an instrument to evaluate PC in children and how it is being applied. In view of the obtained results, accelerometry is being used in children mainly to assess static PC and PC during the gait, as well as to identify the differences between healthy children and children with developmental disorders. Its degree of implementation could be considered as moderate, taking into account the wide range of databases consulted and the inclusion of samples that were not exclusively composed of children.

In all the analyzed studies, there was great heterogeneity in the procedures and protocols of accelerometric evaluation. That is, there was no consensus on a correct and, consequently, more reliable and valid method for the accelerometric evaluation of PC. Aspects such as the point of anatomical recording varied largely among the different analyzed investigations, although the most frequently used location was the lumbar region [32,34,36,37,39,40,41,43,44,46,47]. In fact, this area, more specifically at the level of the third and fourth lumbar vertebrae, is the most recommended point in the literature as the most reliable ACC recording localization for the quantification of balance [54,55].

The recording frequency used also varied largely, although the most popular was 100–128 Hz [32,33,34,36,37,39,41,43,44], with one study using an even higher frequency [42]. This phenomenon contradicts previous studies on the evaluation of PC using ACCs, since, for the study of human movement, it is recommended to use recordings at lower frequencies (20–50 Hz) or to post-process the data with a filter that removes the possible noise of the signal if higher frequencies are employed [14].

The studied population ranges from healthy children to children with pathologies, such CP [32,34,36,40,42,43,44], DCD [33,38,48,56], and vestibular system pathology [35]. In all the cases, the results were positive, showing significant correlations and findings in combination with other noninstrumental clinical balance tests [34,36,44] and with the results obtained using a strength platform [42], which indicates that ACCs represent a valid and reliable tool for the evaluation of PC.

Separately, several studies were focused on comparing the accelerometric results of healthy children with those of children with pathologies [33,34,37,40,42,45,48], on the exclusive evaluation of healthy children to assess balance throughout childhood, and on the evaluation of ACCs as tools for the assessment of PC [31,39,41,46,47], with all of them showing positive results. The existence of studies that have used healthy samples and samples with pathologies, which involve an alteration of motor control, along with the detection of significant differences between healthy children of different ages, confirms the high sensitivity of this instrument [46,47].

On the other hand, it is not possible to point out which evaluation tests are the most adequate to extract a representative accelerometric measurement of PC, due to the large variety of evaluation procedures followed by the analyzed investigations, among which static balance tests (on monopodal and bipodal support and with eyes open or closed) [42,43,44,45,46,47] and gait tests between six steps and 20 m [32,35,36,37,39,40,46] were the most frequently used. However, this review confirms the clinical possibilities of this instrument for the clinical assessment of PC in pediatrics. If accelerometry is applied to evaluate children with spastic or hypotonic CP, vestibular disorders (such as paroxysmal vertigo or cochlear disorders), or DCD with psychomotor disorders and/or brain dysfunction, it is probably possible to (a) get early detection signs of these pathologies and dysfunctions, (b) specifically identify, for each child, their limitations in the development of balance strategies, and consequently, (c) design therapeutic interventions to facilitate and train those undeveloped or less effective strategies for each patient.

We must be cautious when interpreting the results obtained in this systematic review, since the search had limitations, such as the use of six databases. Another aspect to consider is the search equations and terms used among the inclusion criteria, since the use of a different terminology would modify the number of articles found. However, this review followed the methodology and criteria used in other reviews [14,29]. At the same time, it should be noted that the studies analyzed do not have methodologies with a high level of evidence: most of the articles found consist of observational studies, with a small sample size, not randomized or controlled, and do not compare the results obtained by the ACCs with the gold standard for evaluating PC as force platforms.

## 5. Conclusions

ACCs are instruments that provide reliable information about PC in a more sensitive manner than functional test sets traditionally used in the clinical scope. Accelerometry has a discrete degree of implementation as an evaluation tool to assess PC and its method has not been protocolized and standardized yet.

Further research of higher methodological quality is necessary to define a systematic method for the evaluation of PC in pediatrics, in order to delve into the development of this capacity and its alterations in different neurodevelopmental disorders.

## Figures and Tables

**Figure 1 diagnostics-11-00008-f001:**
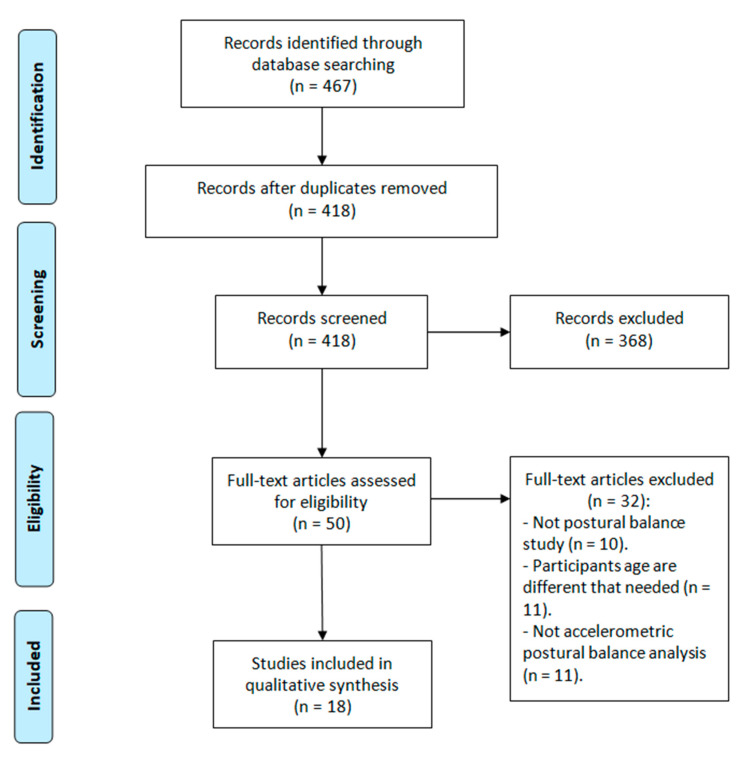
Preferred Reporting Items for Systematic Reviews and Meta-analyses (PRISMA) flow diagram.

**Table 1 diagnostics-11-00008-t001:** Methodological characteristics of the articles reviewed.

Characteristic	Brett et al. (2018) [31]	Cheng et al. (2019) [38]	Crossley et al. (2018) [39]
**Objective**	To provide normative data on static balance using the Sway Balance System method.	To explore the effectiveness of neuromuscular training in improving static balance.	To investigate the speed and angle of turn during gait.
**Study design**	Descriptive cross-sectional.	Randomized clinical trial.	Descriptive cross-sectional.
**Population**	Children and young adults with TD.	DCD children and TD children as a control group.	TD children.
**Sample size**	3763 (1216 women: 32%).	88 (44 control group) (44 girls: 50%).	19 (9 girls: 47%).
**Age** (mean ± standard deviation)	16.3 ± 2.6 years (range: 9–21).	7.6 ± 1.2 years (range: 6–9).	10.1 ± 0.5 years (range: 9–12).
**Motor tests included**	Balance Error Score System Reaction time.	Computerized dynamic posturography.	Three minutes walking at four different speeds and turning at 4 different angles (0º, 45º, 90º, and 180º).
**Accelerometer used**	Triaxial accelerometer iOS (iPad, iPhone and iPod Touch, Apple Inc., Cupertino, CA, USA).	Triaxial accelerometer (Biometrics, Newport, UK).	Triaxial accelerometer (SLAM Tracker, Wildbyte Technologies Ltd., Swansea, UK).
**Frequency of data collection**	*Not specified.*	*Not specified.*	100 Hz.
**Sensor placement**	Hand	Sensors of computerized dynamic posturography.	2 accelerometers: right iliac crest and in the center of the lumbar area.
**Variables analyzed**	Magnitude of accelerations. Reaction time (after hearing a sound, the time it took to bring the device to their chest).	Magnitude of accelerations. Latency time (time between the first detected acceleration and the time of muscle activation of the leg by EMG)	Dynamic vector acceleration of the body.
**Comparison with gold standard or others**	Balance Error Score System	No.	No.

DCD: developmental coordination disorder; EMG: electromyography; TD: typical development.

**Table 2 diagnostics-11-00008-t002:** Methodological characteristics of the articles reviewed.

Characteristic	Fong et al. (2015) [45]	García-Liñeira et al. (2020) [46]	García-Soidán et al. (2020) [47]
**Objective**	To compare neuromuscular performance, balance, and motor skills performance scores in children with DCD and TD.	To determine the reliability and internal consistency of gait and static balance measurement with accelerometry.	To evaluate the effect of the practice of physical activity during childhood on the development of postural control.
**Study design**	Exploratory cross-sectional.	Descriptive cross-sectional.	Descriptive cross-sectional.
**Population**	DCD and TD children.	TD children.	TD children.
**Sample size**	247 (DCD children: 117) (84 girls: 34%).	70 (35 girls: 50%).	118 (54 girls: 45.8%)
**Age** (mean ± standard deviation)	7.5 ± 1.4 years (range: 6–10)	9 ± 1.8 years (range: 6–12).	10.3 ± 1.2 years (range: 8–12)
**Motor tests included**	Static test in standing position receiving a push at T12 from behind.	Single leg support with eyes open and closed and single leg support on a mat for 30 s. 20 m walk test.	20 m walking test. Single-leg static balance test.
**Accelerometer used**	Triaxial accelerometer ACL300 (Biometrics, Newport, UK).	Triaxial accelerometer GT3X+ (Actigraph, Pensacola, FL, USA).	Triaxial accelerometer GT3X+ (Actigraph, Pensacola, USA).
**Frequency of data collection**	*Not specified.*	50 Hz and a Threshold filter.	50 Hz.
**Sensor placement**	Sternum.	L4.	L4.
**Variables analyzed**	Acceleration in the antero-posterior axis.	Mean and maximum of three axes and root mean square of them.	Mean and maximum of three axes and root mean square of them.
**Comparison with gold standard or others**	Movement Assessment Battery for Children.	No.	No.

DCD: developmental coordination disorder; TD: typical development.

**Table 3 diagnostics-11-00008-t003:** Methodological characteristics of the articles reviewed.

	Iosa et al. (2018) [40]	Jobbágy et al. (2016) [48]	Kim et al. (2018) [43]
**Objective**	To assess trunk accelerations when walking in children with CP.	To evaluate postural control with the use of the Huple device and accelerometers.	To determine the relationship between trunk control and function of the upper limbs.
**Study design**	Blinded and randomized clinical trial.	Descriptive cross-sectional.	Descriptive cross-sectional.
**Population**	CP and TD children.	Children with different birth injuries (hypotonia, maturational delays, motor pathologies, and brain dysfunction).	Children with CP (with diplegia or hemiplegia).
**Sample size**	24 (CP children: 12) (10 girls: 42%).	10 (5 girls: 50%)	15 (5 girls: 33%).
**Age** (mean ± standard deviation)	Cerebral palsy: 5.7 ± 2.3 years (range: *not specified*).	*Not specified* (range: 3–8).	9 ± 1.1 years (range: 7–13).
**Motor tests included**	10 m walk test.	One- and two-dimensional digital games controlled by the Huple.	Static test in standing position. Trying to reach an object while sitting.
**Accelerometer used**	Inertial measurement units MIMUs, Opal (APDM Inc., Oregon, USA).	Inertial measurement units x-IMU (x-io Technologies Limited, Bristol, UK).	Triaxial accelerometer (Fitmeter; Fit.Life Inc., Suwon, Korea).
**Frequency of data collection**	*Frequency not specified.* 4th-order low pass Butterworth filter (20 Hz).	64 Hz.	128 Hz.
**Sensor placement**	3 devices: head (occipital), sternum and pelvis (between sacrum and L5).	On Huple.	L3 and 3rd metatarsal.
**Variables analyzed**	Three axes accelerations, angular velocity, and magnetic field vector components	*Not specified.*	Antero-posterior and medio-lateral accelerations.
**Comparison with gold standard or others**	No.	No.	Quality of Upper Extremity Skills Test, Jebsen Taylor Hand Function Test, Box and Blocks Rest, Abilhand-KIDS questionnaire.

CP: cerebral palsy; TD: Typical development.

**Table 4 diagnostics-11-00008-t004:** Methodological characteristics of the articles reviewed.

Characteristic	Kim et al. (2018) [44]	Linder et al. (2018) [41]	Mutoh et al. (2016) [32]
**Objective**	To measure trunk swing in sitting and standing position.	To apply the Cleveland Clinic Postural Stability for the evaluation of static balance	To assess the effect of hippotherapy on the move in subjects with CP.
**Study design**	Descriptive cross-sectional.	Descriptive cross-sectional.	Case series
**Population**	CP children.	TD children.	Bilateral spastic CP children.
**Sample size**	15 (5 girls: 33%).	70 (35 girls: 50%).	3 (1 girls: 33%).
**Age** (mean ± standard deviation)	9 ± 1.1 years (range: 7–13).	9 ± 1.8 years (range: 6–12).	10.1 years (range: *not specified*).
**Motor tests included**	Static balance in standing and sitting.	Balance Error Scoring System.	10 m walk test.
**Accelerometer used**	Triaxial accelerometer Fitmeter (Fit.Life Inco., Suwon, Korea).	Triaxial accelerometer ST Micro LIS33IDLH from Ipad (Apple, Inc., Cupertino, CA, USA).	Triaxial accelerometer MG-M1110-HW (LSI Medience, Tokyo, Japan).
**Frequency of data collection**	128 Hz.	100 Hz and low pass Butterworth filter (3.5 Hz).	100 Hz.
**Sensor placement**	L3.	Sacrum (iliac crest level).	L3.
**Variables analyzed**	Three axes mean accelerations.	Percentile ranking for the three axes accelerations.	Step rate, step length, gait speed, mean acceleration, and horizontal/vertical displacement ratio.
**Comparison with gold standard or others**	Trunk Impairment Scale and Trunk Control Measurement Scale.	Balance Error Score System	No.

CP: cerebral palsy; TD: Typical development.

**Table 5 diagnostics-11-00008-t005:** Methodological characteristics of the articles reviewed.

Characteristic	Saether et al. (2015) [36]	Shiratori et al. (2016) [42]	Speedtsberg et al. (2018) [33]
**Objective**	To evaluate the trunk control relationship between seated and during gait	To assess anticipatory postural adjustments in static equilibrium when supporting a load with the hands.	To investigate trunk stability during treadmill walking.
**Study design**	Descriptive cross-sectional.	Descriptive cross-sectional.	Descriptive cross-sectional.
**Population**	Children with CP spastic.	Spastic CP with diplegia or hemiplegia and TD children.	Children with DCD and TD.
**Sample size**	26 (9 girls: 34.6%)	27 (9 control) (19 girls: 70%).	18 (10 control) (5 girls: 28%).
**Age** (mean ± standard deviation)	13.5 ± 3 years (range: 8–18)	11.8 ± 1.8 years (range: 7–17).	9 ± 1.5 years (range: *not specified*).
**Motor tests included**	5 m walking test.	Stand and support a falling load with their hands.	4 min walking on a treadmill at normal speed.
**Accelerometer used**	Triaxial accelerometer MTx (XSens, Enschede, The Netherlands).	Uniaxial accelerometer (PCB Piezotronics, Depew, NY, USA).	Triaxial accelerometer MQ16 (Marq Medical, Farum, Denmark).
**Frequency of data collection**	100 Hz.	1000 Hz and 2nd-order low pass Butterworth filter (20 Hz) with zero phase shift.	256 Hz.
**Sensor placement**	L3.	In dominant hand.	Sternum.
**Variables analyzed**	Mean, regularity and root mean square of three axes accelerations. Number and duration of steps.	Peak acceleration.	Short term local dynamic stability (λs), root mean square and relative root mean square.
**Comparison with gold standard or others**	Trunk Impairment Scale and Trunk Control Measurement Scale.	Force platform (OR-6, AMTI, Watertown, MA, USA) with Labview software (National Instruments, Austin, TX, USA).	No.

CP: cerebral palsy; DCD: developmental coordination disorder; TD: typical development.

**Table 6 diagnostics-11-00008-t006:** Methodological characteristics of the articles reviewed.

Characteristic	Summa et al. (2015) [37]	Tramontano et al. (2017) [34]	Wolter et al. (2019) [35]
**Objective**	To compare the postural adjustments in children with CP and TD	To determine the effect of doing a simultaneous cognitive task when walking on balance.	To determine the influence of the BalanCi application on the bilateral cochlear implant on balance during gait.
**Study design**	Descriptive cross-sectional.	Descriptive cross-sectional.	Prospective, blinded, case-control.
**Population**	Children with CP and TD.	Children with CP, adults with stroke and controls with TD.	Children with bilateral cochlear implants and vestibular loss.
**Sample size**	40 (CP children: 20) (18 girls: 45%)	85 (50 control) (40 girls: 47%).	26 (10 control) (9 girls: 35%).
**Age** (mean ± standard deviation)	5.8 ± 2.2 years (children range: 2–9).	34.9 ± 4.3 years (children range: 3–12).	14.1 ± 3.6 years (range: 6–17).
**Motor tests included**	10 m walking test.	Children: Pediatric Balance Scale and Gross Motor Function Measure. Adults: Berg Balance Scale and Timed Up & Go.	6 m walk test and Modified Clinical Test Sensory Interaction in Balance.
**Accelerometer used**	Inertial measurement units MIMUs, Opal (APDM Inc., Portland, OR, USA).	Triaxial accelerometer (*unspecified model*).	Ipod Touch (Apple Inc., Cupertino, CA, USA).
**Frequency of data collection**	128 Hz and 4th-order low pass Butterworth filter (20 Hz).	100 Hz and low pass filter (20 Hz).	*Not specified.*
**Sensor placement**	Head, sternum, and sacrum–L5.	L2–L3.	Vertex of the head.
**Variables analyzed**	Root mean square of three axes.	Accelerations in the three axes and root mean square.	Root mean square.
**Comparison with Gold Standard or others**	No.	Pediatric Balance Scale.	No.

CP: cerebral palsy; TD: typical development.

## Data Availability

No new data were created or analyzed in this study. Data sharing is not applicable to this article.

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
