# Peer review of "Accelerometric Assessment of Postural Balance in Children: A Systematic Review"

_diagnostics, 2020, doi:10.3390/diagnostics11010008_

Round 1
Reviewer 1 Report
I recommend some changes:
- The abstract is too general. Please provide more details.
- The research should be registered in PROSPERO website before resubmission.
- The mean difference of analysed outcomes in described articles must be calculated in results section.
- Please add limitations of study at the end of discussion.
Author Response
Dear Editor and Reviewer of Diagnostics:
Thank you very much for your suggestions and contributions to improve the quality of the manuscript. Following your indications, we respond, point by point, to the reviewers' comments.
In the text, all the modified or added sentences have been written in red to facilitate the correction by the reviewers.
- The abstract is too general. Please provide more details.
The authors have expanded the Introduction, better contextualizing the subject of study.
- The research should be registered in PROSPERO website before resubmission.
The authors have completed the registration.
The response time for sending corrections that the Diagnostics Editor has given us has been five days. However, PROSPERO's response time to approve the registration of the review is 10 working days and we have been advised that it can be extended to three months for requests from outside the United Kingdom.
For these reasons, the authors have not been able to add the registration number in the manuscript but we send you as proof the capture of the registration request status in PROSPERO (You can see it in the .docx version of the response letter).
- The mean difference of analysed outcomes in described articles must be calculated in results section.
The mean difference of analyzed outcomes in described articles should not be included since the instruments used were not the same in any case (no two articles have used the same accelerometer, with the same configuration, with the same data processing, during performing the same tests). Therefore, the numerical comparison of results is not adequate.
- Please add limitations of study at the end of discussion.
The authors have expanded the Discussion with the methodological limitations identified.
Once again, thank you very much for the time spent and the interest shown in this work; as well as in the positive evaluations you have given of it.
Receive a warm greeting,
The authors.

Reviewer 2 Report
In the manuscript, the authors substantially show the presentation of the contributions and the experimental results. In my opinion, the version of the manuscript is now acceptable for publication.
Author Response
Dear Editor and Reviewer of Diagnostics:
Thank you very much for your suggestions and contributions to improve the quality of the manuscript. Following your indications, we respond, point by point, to the reviewers' comments.
In the text, all the modified or added sentences have been written in red to facilitate the correction by the reviewers.
- In the manuscript, the authors substantially show the presentation of the contributions and the experimental results. In my opinion, the version of the manuscript is now acceptable for publication.
The authors greatly appreciate the positive evaluation of our work. At the same time, we hope that the modifications made to the manuscript (following the corrections of Reviewer 1) will also be to your liking.
Once again, thank you very much for the time spent and the interest shown in this work; as well as in the positive evaluations you have given of it.
Receive a warm greeting,
The authors.
Reviewer 3 Report
This systemic review article suggested that accelerometry has a discrete degree of implementation as an evaluation tool to assess postural control. This study is well designed. A revision is suggested.
- Please discuss the clinical implications of this study, please focus on specific diseases.
- Please discuss the limitations of this study.
Author Response
Dear Editor and Reviewer of Diagnostics:
Thank you very much for your suggestions and contributions to improve the quality of the manuscript. Following your indications, we respond, point by point, to the reviewers' comments.
In the text, all the modified or added sentences have been written in red to facilitate the correction by the reviewers.
- Please discuss the clinical implications of this study, please focus on specific diseases.
The authors have expanded the Discussion with the practical implications of the use of accelerometers for the evaluation of postural control in pediatrics.
- Please discuss the limitations of this study.
The authors have also expanded the limitations of this research.
Once again, thank you very much for the time spent and the interest shown in this work; as well as in the positive evaluations you have given of it.
Receive a warm greeting,
The authors.
Round 2
Reviewer 1 Report
Unfortunately, the authors did not make all suggested corrections. I recommend to reject the paper in this form, and then register on PROSPERO database and calculate proposed in previous review parameters. Later the paper could be submitted once again. Now, it is not enough.
Author Response
Dear Editor and Reviewer of Diagnostics:
Thank you very much for your suggestions and contributions to improve the quality of the manuscript. Following your indications, we respond, point by point, to the reviewers' comments.
In the text, all the modified or added sentences have been written in red to facilitate the correction by the reviewers.
- Unfortunately, the authors did not make all suggested corrections. I recommend to reject the paper in this form, and then register on PROSPERO database and calculate proposed in previous review parameters. Later the paper could be submitted once again. Now, it is not enough.
The authors are disappointed with this message.
Of the four corrections indicated, the authors have made two as indicated. Regarding the registration in PROSPERO of this systematic review.
The authors enclose in the reply letter (and we repeat below in the PDF version of the answer that the authors have attached) an image that shows that we have made the registration. However, the reviewer would have to know that the response process by PROSPERO is about 10 weeks (and on the web they warn that in the current situation due to COVID-19 it may still be longer). Waiting three months (or more) to be able to publish a systematic review that, according to its corrections, has very few shortcomings is contradictory. Since the revision itself would be obsolete after this waiting period.
At the same time, the authors recognize the importance of registering systematic reviews in PROSPERO and, from now on, it will be an essential step for us each time we initiate a review.
On the other hand, the mean difference of analyzed outcomes in described articles should not be included since the instruments used were not the same in any case (no two articles have used the same accelerometer, with the same configuration, with the same data processing, during performing the same tests). Therefore, the numerical comparison of results is not adequate.
Finally, the authors hope that you will rethink your decision, especially taking into account the positive evaluations received by other reviewers and that, after the corrections made (both by your advice and that of the other reviewers) the quality of the manuscript has improved significantly.
Once again, thank you very much for the time spent and the interest shown in this work; as well as in the positive evaluations you have given of it.
Receive a warm greeting,
The authors.

Round 3
Reviewer 1 Report
I understand the frustration. Unfortunately, my opiniom is the same.